# Teacher-generated pseudo human spatial-attention labels boost contrastive learning models

## Abstract

Human spatial attention conveys information about the regions of scenes that are important for performing visual tasks. Prior work has shown that the spatial distribution of human attention can be leveraged to benefit various supervised vision tasks. Might providing this weak form of supervision be useful for self-supervised representation learning? One reason why this question has not been previously addressed is that self-supervised models require large datasets, and no large dataset exists with ground-truth human attentional labels. We therefore construct an auxiliary teacher model to predict human attention, trained on a relatively small labeled dataset. This human-attention model allows us to provide an image (pseudo) attention labels for ImageNet. We then train a model with a primary contrastive objective; to this standard configuration, we add a simple output head trained to predict the attentional map for each image. We measured the quality of learned representations by evaluating classification performance from the frozen learned embeddings. We find that our approach improves accuracy of the contrastive models on ImageNet and its attentional map readout aligns better with human attention compared to vanilla contrastive learning models.

## 1 Introduction

Deep learning models have made significant progress and obtained notable success on various vision tasks. Despite these promising results, in many applications humans continue to perform better than deep learning models. A notable reason is that deep learning models have a tendency to learn "short-cuts", i.e., giving significance to physically meaningless patterns or exploiting features which are predictive in some settings, but not causal [12]. Examples include focusing on less significant features such as background and textures [8]. These models yield representations that are less generalizable and lead to models that are highly sensitive to small pixel modulations [22].

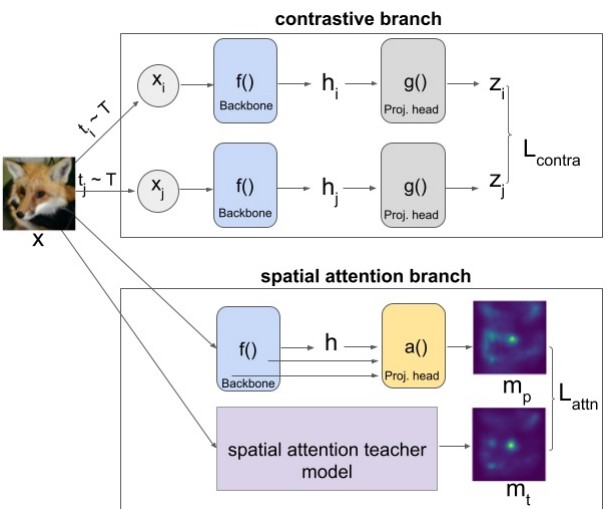

Figure 1: Illustration of the proposed method of aligning model spatial attention to humans attention using a teacher auxiliary model.

Human vision on the other hand is known to be much more robust and generalizable. One major difference between human and machine vision is that humans tend to focus on specific regions in visual scene [24]. These locations often reflect regions salient or useful to perform a specific vision

Submitted to 4th Workshop on Shared Visual Representations in Human and Machine Visual Intelligence (SVRHM) at NeurIPS 2022. Do not distribute.

task. Machines, instead, initially place equal significance to all regions. A natural question is: will it be beneficial if machine vision models is guided by human spatial attention?

Human spatial attention has been shown to benefit computer vision models in supervised tasks, such as classification [17]. Yet, it is still a question whether adding a form of weak supervision in the form of human spatial attention could similarly benefit self-supervised models. Self-supervised models typically need a large amount of data to yield good representations. To test if training weakly supervised models with human spatial attention cues, we will need to collect a large volume of human spatial attention labels, which is a very expensive process that requires either using trackers to record eye movements [32, 4, 23] or asking humans to highlight regions that they attend to [15, 16]. This process is prohibitively tedious and costly for datasets with millions of examples.

In this work, we explore utilizing human spatial attention as a form of weakly-supervised representation learning for models trained with a contrastive objective. Inspired by knowledge distillation and self-training ideas using teacher models [26, 28], we address the challenge of obtaining spatial attention labels on large scale image datasets by using machine pseudo-labeling. We train a teacher model on a set of limited ground truth human spatial attention labels. We then use this teacher model to generate spatial attention pseudo-labels for the larger ImageNet benchmark. We are then able to utilize the generated spatial attention maps in the contrastive models, and discover that this approach yields representation highly predictive of human spatial attention. Further, we find that the learned representations are better as measured by higher accuracy of the ImageNet classification downstream task. More interestingly, we find that the gains from using teacher models to provide pseudo labels are larger than using the limited ground truth human labels directly when training contrastive models.

## 2   Related work

**Contrastive learning:** Contrastive learning has gained popularity in the past few years for self-supervised and semi-supervised representation learning. In general, contrastive learning aims to learn similar representations for similar data pairs and different representations for different pairs. SimCLR [5] utilized MLP projection heads and strong data augmentation for constructing similar pairs and demonstrated great gains in image classification downstream tasks. Zbontar et al. [30] used a different formulation by encouraging the empirical cross correlation of the representations of two versions of augmented mini-batch to be close to identity. He et al. [10] further proposed building large dictionaries for self-supervised learning (MOCO), and Chen et al. [6] achieved better results on image classification and object detection tasks when combining advances from SimCLR and MOCO.

**Human spatial attention data collection:** Human visual system has developed an attention mechanism that focuses on regions in the visual space that are of interest or highly informative to the vision task ([7, 27]). Eye trackers are often used to collect human spatial attention [32, 4, 23]. Many gaze data sets ([2]) have already been collected with these eye trackers, where users are either asked to view the image/video freely, or conduct specific tasks like classification or object detection. Besides eye trackers, human spatial attention data can also be collected via mouse tracking [15, 16], e.g., users see a blurry version of an image, and then click on regions they want to see more clearly, mimicking human's peripheral vision based on neurophysiological and psychophysical studies [11, 16]. Salicon [15] dataset is one of the largest spatial attention datasets, contains around 20K images, each with attention labels from 50-60 participants, via a mouse tracking system, under free viewing setting . Yet, this data is still orders of magnitude smaller than those needed to train self-supervised models.

**Spatial attention of computer vision models:** Spatial attention in neural networks can be mainly categorized into post-hoc attention like class activation map (CAM) ([31]), and trainable attention (e.g., Wang et al. [25], Jetley et al. [13], Guo et al. [9]). Post-hoc spatial attention methods have been proposed to estimate regions in the image that are important or give rise to model decisions, often for model interpretation. In supervised settings where classification labels are known, the simplest and most direct method is class activation map (CAM) [[31]]. CAM uses class labels to extract the feature map that is most informative about the true class of an image. Grad-CAM [21] generalizes the CAM to apply to any model with any downstream task. ContraCAM [18] applies Grad-CAM assuming downstream task of contrastive learning, thus allowing computing spatial attention maps with no class label supervision. Mo et al. [18] proposed to utilize the spatial attention information learned from ContraCAM to design data augmentation strategies to discourage contextual and background biases in a scene. Yet, those augmentation are complex and ad-hoc. Here we propose an end-to-end framework to predict spatial attention targets rather than using spatial attention to design augmentation policies. Lai et al. [17] conducted experiments to use human spatial attention

to supervise model attention, for three tasks (salient object segmentation, video action recognition and fine-grained image classification) and demonstrated that human spatial attention is beneficial. However, it still remains a question whether such benefits could be extended to contrastive learning.

**Teacher model pseudo-labeling:** Previous work on knowledge distillation and machine self training has demonstrated that machine teaching machines approaches may address the challenge of labeling large datasets. In image classification, Xie et al. [28] demonstrated that training a model to classify images then use that model to provide pseudo-labels for larger data improved classification performance. Similar idea is applied in West et al. [26] for language models. Inspired by these successes, we train a teacher model on smaller human attention data and use this model to generate new spatial attention pseudo labels for ImageNet benchmark (see Figure 1).

## 3 Methods

We first train a spatial attention teacher model on Salicon data [15], then use the teacher-generated attention to predict pseudo human attention labels from ImageNet dataset. We then use the pseudo attention labels as targets of the contrastive learning's spatial-attention training objective. Our teacher model follows the architecture of [19], but is simplified with less channels/layers. Whereas existing attention prediction models [14, 19, 1] finetune pretrained classification backbones, we instead use randomly initialized backbone to avoid any leak of class label information

As shown in Figure 1, the proposed model consists of two branches: the contrastive branch and the spatial attention branch. The contrastive branch is the same as the original SimCLR method, which applies augmentations to image $x$ to get different variants $x_i$ and $x_j$, and learns the representation $h_i$ and $h_j$ via a feature extractor backbone network (we use ResNet-50 ), then use a projection head to map $h_i/h_j$ to $z_i/z_j$, where the contrastive loss is applied.

For the spatial-attention branch, we apply a global average pooling to the intermediate outputs of the model backbone, i.e. the last three blocks of the Resnet backbone, including both low level and high level features. Then we select the intermediate representations corresponding to the max channel and resize with bilinear interpolation to the image resolution. Finally we stack the representations together, pass them into a linear readout layer, and use the output as our final spatial attention prediction $m_p$.

We use the pseudo labels from the teacher model as target spatial attention $m_t$, and then optimize the network to bring spatial attention output $m_p$ close to $m_t$ using KL divergence loss. In order to cover more human attention details, we also generate pseudo fixation points from the pseudo attention maps [1] and use normalized scanpath saliency (NSS) loss [3] as an additional loss. We hypothesize that this method regularizes the training of the feature extractor backbone rather than explicitly enforce the network to generate masked representations that match the spatial attention maps. Note that for attention branch, there is no augmentation applied to each image $x$, since human attention is not invariant to transformation (e.g., a human looking at a cropped image may attend to different region compared to a consistent crop of human attention map of the original image).

## 4 Results

### 4.1 Spatial attention guided models are highly predictive of human attention

In this section, we explore whether the use of auxiliary teacher model to provide spatial attention pseudo labels on ImageNet better aligns contrastive model's attention with human attention. We define model spatial attention here as the ability to predict spatial attention mask from the model backbone features by a simple readout layer proposed in Section 3.

We then trained two ResNet-50 backbones using the SimCLR objective from Chen et al. [5]. We added additional attention losses as discussed in Section 3. For the first model (baseline), we placed a stop gradient operation between the backbone features and the attention projection head to prevent attention information from informing the learned features, whereas for the second model (attention guided) we allowed the learned attention gradients to flow back to the backbone.

We evaluated the degree the predicted attention maps is aligned with human attention by computing the Pearson's correlation between the model predicted attention and a human attention dataset [23]

---

[1]We first extract the point with highest value in current attention map, then generate a new attention map by subtracting a Gaussian blur around the extracted point from the current attention map. The process is repeated with the new attention map until the maximal value of the attention map is smaller than a threshold.

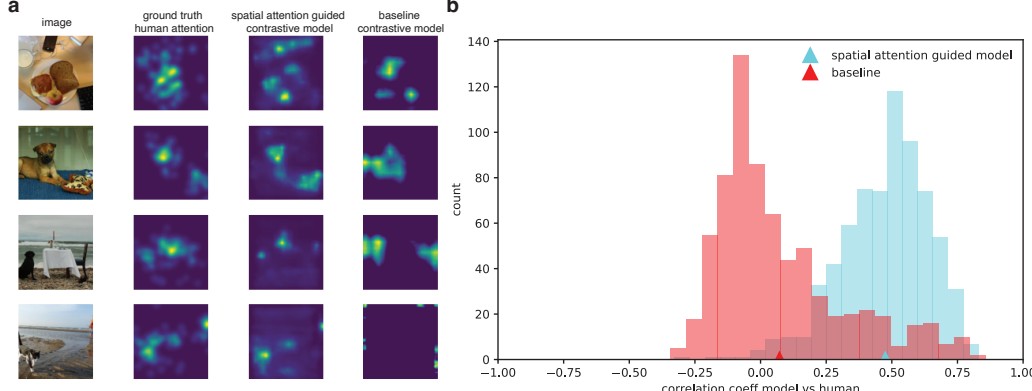

Figure 2: a) Examples comparing spatial attention maps predicted by different models vs ground truth human attention data on OSIE dataset [23]. b) Distribution of correlation coefficients between attention maps predicted by models vs ground truth human maps on OSIE dataset [23].

collected by mobile eye tracker [23] on OSIE images[29]. Our results are summarized in Figure 2. We find that the baseline model is positively correlated with human attention (ttest: $\rho = 0.07$ $p < 0.001$) suggesting that the contrastive loss produces features that are predictive of human attention to some extent. Yet, the correlation was generally close to 0 and explains only $0.5\%$ of data variance. The correlation of the attention guided model with human attention is much stronger (ttest: $\rho = 0.48$ $p < 0.001$) than the baseline model (See Fig 2a for qualitative examples and Fig 2b for quantitative analysis. Two samples ttest: $p < 0.001$), and thus more faithfully reflecting human visual attention. [2]

## 4.2 Spatial attention guided models are more accurate than baselines

We evaluate the quality of the representations learned by spatial attention guidance framework using the typical contrastive learning evaluation criteria: fitting an ImageNet [20] linear classifier on top of the frozen representation (in practice we place stop gradient at the end of the backbone and train the classifier concurrently while training the backbone). We compute Top 1 accuracy on ImageNet validation set and compare the results with baselines. As shown in Table 1, we observe around 0.6% accuracy gain on ImageNet compared to vanilla SimCLR. We further explore an alternative way of in-

Table 1: ImageNet Top-1 classification accuracy for different models (mean $\pm$ SE for 3 seeds, except for * 1 seed).

| Model | Accuracy (%) |
|---|---|
| Contrastive | $67.61 \pm 0.04$ |
| Contrastive attn. teacher | $\mathbf{68.23 \pm 0.08}$ |
| Contrastive attn. augmented | $56.35^*$ |
| Supervised | $75.91 \pm 0.10$ |
| Supervised attn. teacher | $76.08 \pm 0.03$ |

corporating human attention data. Rather than using pseudo attention labels on ImageNet from the teacher model, we add Salicon data to to the training data, and directly predict attention labels from Salicon data (though we use a different readout layer consists of convolution and transpose convolution layers instead of the simple linear layer). Interestingly, we find this method to lead to worse performance.

To investigate whether the spatial attention guidance framework benefits supervised models in the same way, we conducted the same experiments for supervised models. Supervised models similarly benefit from this framework, yet the gain is limited compared to the contrastive models perhaps due to the higher accuracy the supervised model achieves compared to the contrastive model.

## 5 Conclusion

In this work, we explored using human spatial attention to aid training of contrastive learning models. We overcome the challenge of obtaining attention labels for large dataset by utilizing a teacher model trained on limited ground truth human attention labels to provide pesudo-attention labels for ImageNet. Our results demonstrate that contrastive models trained with those pseudo-attention labels become more predictive of human attention and we obtain better representations.

---

[2]Note that the teacher model is trained on Salicon data with human attention ground-truth collected via mouse tracking [15], while the evaluation data set is OSIE image set with attention data collected directly from mobile eye tracker [23], thus it more faithfully represent human spatial attention

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
