# OpenReview forum: "Teacher-generated pseudo human spatial-attention labels boost contrastive learning models"
_NeurIPS.cc/2022/Workshop/SVRHM — SVRHM Poster_

### Official Review · Reviewer_4zrT · 2022-10-01
**Lack of experiments and comparison**

**Rating:** 4
**Confidence:** 4

**Review:**

Summary: This work tries to incorporate supervision in the form of human spatial attention into the framework of contrastive learning.

It is a good attempt to incorporate human spatial attention information into contrastive learning. However, the experiments are not conducted thoroughly which makes the conclusion not conclusive.

Comments:
1. Only one dataset is used for the experiments. If it is hard to find datasets with ground-truth human spatial attention maps, it is still possible to study whether the learned teacher model can be transferred to other datasets.
2. It would be clearer how spatial attention supervision can boost the performance of contrastive learning if you compare the results of different contrastive learning methods. In this work, only SimCLR is considered.
3. If the proposed method uses one form of supervisory information, it should also be compared with others in the same or different forms.

---

### Official Review · Reviewer_tfBr · 2022-10-14
**Very interesting approach, with promising results**

**Rating:** 8
**Confidence:** 4

**Review:**

This paper incorporates human spatial attention data into a contrastive learning model, resulting in a major improvement in the model's ability to predict human gaze position data and a modest improvement in the core computer vision task of classifying Imagenet images. The paper is well-written: both the model architecture and the results are easy to understand from the figures and writing.

Below, I offer a few suggestions / comments, which I hope will be helpful for strengthening this very interesting paper!

  - To address the issue that collecting human spatial attention gaze data is costly and difficult, the authors cleverly exploit a "teacher" model, which is trained to predict human gaze positions from the Salicon saliency dataset and then subsequently used to generate saliency maps for training the attention branch of the authors' contrastive learning model. However, this necessarily means that the performance of their new model in predicting attention maps is bounded by the performance of the teacher model. I would be curious to see, in figure 2, how their model stacks up to the upper bound performance of the teacher model, relative to the ground truth data.
  - Relatedly, I don't entirely understand why the spatial attention maps are evaluated on a different dataset (OSIE) than the one which the teacher model was trained on (Salicon).
  - The authors briefly mention that they are also using pseudo fixation points to compute an additional loss term based on normalized scanpath saliency loss and hypothesize that this scanpath-based loss acts as a regularizer. I am curious as to how important this turns out to be in the accuracy of the model. Would the model have done as well without this loss component?